# Long-Term Outcome of Indomethacin Treatment in Pediatric Patients with Paroxysmal Hemicrania—A Case Series

**DOI:** 10.3390/children8020101

**Published:** 2021-02-03

**Authors:** Maximilian David Mauritz, Anna Enninger, Christine Wamsler, Julia Wager, Boris Zernikow

**Affiliations:** 1German Paediatric Pain Centre, Children′s and Adolescents′ Hospital, Witten/Herdecke University, 45711 Datteln, Germany; c.wamsler@kinderklinik-datteln.de (C.W.); j.wager@deutsches-kinderschmerzzentrum.de (J.W.); b.zernikow@kinderklinik-datteln.de (B.Z.); 2Department of Children′s Pain Therapy and Paediatric Palliative Care, Faculty of Health, School of Medicine, Witten/Herdecke University, 58448 Witten, Germany; anna.enninger@uni-wh.de; 3Centre for Rare Pain Disorders in Young People, Centre for Rare Diseases Ruhr (CeSER), 45711 Datteln, Germany

**Keywords:** headache, pain, trigeminal autonomic cephalalgias, paroxysmal hemicrania, children, indomethacin

## Abstract

Paroxysmal Hemicrania is a rare form of primary headache in children and adolescents, belonging to the group of trigeminal autonomic cephalalgias. Patients suffer from severe, short-lasting unilateral headaches accompanied by symptoms of the autonomic system on the same side of the head. The short duration of attacks distinguishes Paroxysmal Hemicrania from other trigeminal autonomic cephalalgias. Indomethacin is the treatment of choice, and its effectiveness provides a unique diagnostic criterion. However, the long-term outcomes in children are highly underreported. In this case-series, *n* = 8 patients diagnosed with Paroxysmal Hemicrania were contacted via telephone 3.1 to 10.7 years after initial presentation. A standardized interview was conducted. *n* = 6 patients were headache-free and no longer took indomethacin for 5.4 ± 3.4 years. The mean treatment period in these patients was 2.2 ± 1.9 years. Weaning attempts were undertaken after 1.7 ± 1.3 months; in *n* = 3 patients, more than one weaning attempt was necessary. *n* = 2 patients were still taking indomethacin (4.5 and 4.9 years, respectively). Both unsuccessfully tried to reduce the indomethacin treatment (two and six times, respectively). Adverse effects appeared in *n* = 6 (75%) patients and led to a discontinuation of therapy in *n* = 2 patients. Our long-term follow-up suggests that in a substantial proportion of pediatric patients, discontinuing indomethacin therapy is possible without the recurrence of Paroxysmal Hemicrania.

## 1. Introduction

Trigeminal autonomic cephalalgias (TAC) are a group of rare primary headache disorders. They are characterized by sudden unilateral headaches in the trigeminal nerve region, accompanied by symptoms of the autonomic nervous system on the same side of the head. The International Classification of Headache Disorders, 3rd edition (ICHD-3) [1] distinguishes Cluster Headache, Paroxysmal Hemicrania, SUNCT syndrome (Short-lasting unilateral neuralgiform headache with conjunctival injection and tearing), SUNA syndrome (Short-lasting unilateral neuralgiform headache attacks with cranial autonomic symptoms), and Hemicrania Continua.

Paroxysmal Hemicrania (PH) is a severe unilateral short-lasting headache, occurring more than five times a day (see Table 1 for the ICHD-3 diagnostic criteria). Patients describe the pain as sharp, stabbing, or agonizing, usually localized around the eye [2]. The short duration of 2–30 min distinguishes PH from other TACs.

Attacks present in association with at least one ipsilateral autonomic symptom, such as ptosis, miosis, eye tearing, conjunctival injection, nasal obstruction, or rhinorrhea. Contrary to adult patients, restlessness or agitation has to date not been described in pediatric case reports [3]. In addition to chronic PH, there are also episodic PH cases, in which frequent pain attacks are followed by pain-free intervals of at least three months [1].

Typically, PH occurs in adults, but there are some reports of children also being affected [2,3,4,5]. One core diagnostic criterion is a good response to indomethacin, which is also the treatment of choice [6,7,8]. The long-term outcomes of indomethacin treatment in children are highly underreported and therefore enormously important to understand. It remains unclear for the pediatric patients and their parents whether the indomethacin treatment can be tapered at one point or whether a lifelong medication is crucial.

The aim of the current study was to provide long-term follow-up data of children and adolescents diagnosed with PH. Specifically, we aimed to report on the potential for the successful discontinuation of indomethacin treatment.

## 2. Materials and Methods

We performed a retrospective study of all patients presenting to a tertiary pain center between 2003 and 2017 with PH with a follow-up of at least three years. Patients and parents were contacted via telephone and informed about the study. If they agreed to participate, a standardized interview was conducted about the disease’s course and its association to indomethacin therapy. Additionally, medical case files were reviewed.

Descriptive statistics such as means, median, standard deviations, and frequencies were computed for the relevant outcome measures. The analyses were performed using SPSS Version 27.0 (SPSS Inc., Chicago, IL, USA). Ethical approval (2017/08/02 BZ3) was provided by the ethical committee of the children’s hospital.

## 3. Results

Between 2003 and 2017, *n* = 16 patients had a suspected or differential diagnosis of PH at the German Paediatric Pain Centre, Children’s and Adolescents’ Hospital Datteln-Witten/Herdecke University. *n* = 12 patients agreed to participate in the telephone interview. *n* = 4 patients were excluded because their suspected diagnosis of PH was not (or no longer) applicable, i.e., only patients with a confirmed PH diagnosis were included.

The remaining *n* = 8 patients with an established diagnosis of PH had a mean age of 8.1 years (range 4.1–11.8; median = 7.6 years) at the time of diagnosis. Their follow-up interviews were conducted 3.1 to 10.7 years after initial presentation. The time from the occurrence of the first PH headache attack to the diagnosis and the initial treatment with indomethacin was 10.5 ± 10.8 months (range 2.5–34.5; median = 10.0 months). The individual patient characteristics are summarized in Table 2. As a necessary condition for the diagnosis, indomethacin therapy was initially effective (no pain attacks) in all patients. The applicability of the ICHD-3 criteria in the study patients is reported in Table 3.

All patients took indomethacin daily. The mean starting doses were 2.75 ± 0.66 mg/kgbw (milligram per kilogram body weight), with a range of 1.3–3.4 mg/kgbw. At the time of the telephone interview, *n* = 6 patients (75%) were headache-free and no longer took indomethacin. Those patients received indomethacin for a mean treatment period of 2.2 ± 1.9 years (range: 0.4–5.7; median = 1.8 years). During their treatment course, indomethacin was reduced stepwise and discontinued without the reoccurrence of headache attacks. Weaning attempts were tried after a mean treatment time of 1.7 ± 1.3 months (range 0.7–4.7; median = 1.1 months).

The tapering period lasted a mean of 2.1 ± 1.9 years per patient (range: 0.2–4.6; median = 1.6 years). In *n* = 3 patients, more than one weaning attempt was necessary since the PH reoccurred during the tapering time. A mean of 2.7 weaning attempts were tried (range: 1–2 unsuccessful tapering trials). *n* = 3 patients had to restart indomethacin after they ceased it. Figure 1 displays the course of indomethacin treatment for all patients who successfully discontinued medication. At the time of the telephone interview, patients were symptom-free for a mean of 5.4 ± 3.4 years (range: 1.4–10.2; median = 5.0 years). In *n* = 2 patients, the indomethacin treatment was stopped because of severe side effects. After interrupting indomethacin treatment, headache reoccurred in both patients. Therefore, the treatment was changed to topiramate therapy with an effective dose of 0.45 and 0.5 mg/kgbw, respectively. The topiramate therapy itself was terminated after two years without recurrence of the headache.

At the time of the telephone interview, *n* = 2 patients were still taking indomethacin (4.5 and 4.9 years, respectively). Both unsuccessfully tried to reduce the application (two and six times, respectively). At all times, the headache attacks reoccurred after days or up to four months later in the patient with episodic PH or even after 36 h in the patient suffering from chronic PH. At the time of the interview, the patients were still on indomethacin therapy and pain-free. Their actual daily indomethacin doses were 0.63 mg/kgbw and 0.27 mg/kgbw, respectively. Figure 2 displays the course of indomethacin treatment for the two patients still receiving indomethacin.

The patients included in this study reported several side effects of indomethacin: gastrointestinal problems like abdominal pain, diarrhea, and constipation (*n* = 4), edema (*n* = 1), and changes in behavior and agitation (*n* = 2). Two patients required proton pump-inhibitor therapy for dyspepsia and gastric ulcers; as reported above, two patients had to discontinue indomethacin therapy because of side effects. No patient experienced severe adverse effects as defined by death, a life-threatening adverse event, inpatient hospitalization or the prolongation of an existing hospitalization, a persistent or significant incapacity, or a substantial disruption of the ability to conduct normal function. Neither of the two patients still receiving indomethacin treatment developed tachyphylaxis.

## 4. Discussion

Our understanding of the pathophysiology of PH—based mainly on functional neuroimaging studies and experimental animal models—remains incomplete [9,10]. The shared feature of all TACs is the activation of the trigeminal-autonomic reflex. This activation, in turn, leads to an increased cranial parasympathetic outflow. The afferent loop of this cascade involves the trigeminal nerve, mainly the fibers of the first division. This afferent signal is apparently responsible for the pain. Sensory neurons in the trigeminal ganglion innervate both intra- and extracranial structures, as well as the meningeal vessels. Nociceptive signals from here reach the trigeminal nucleus caudalis, from where they are transmitted to the thalamus, brainstem, and, in the case of PH, the contralateral posterior hypothalamus. These signals open a reflex arc in the parasympathetic superior salivatory nucleus, which is located in the pons. From here, parasympathetic nerve fibers lead to the sphenopalatine ganglion, which innervates both cerebral vessels as well as the lacrimal and nasal mucosa glands. These parasympathetic circuits and a sympathetic deficit caused by a carotid swelling compressing sympathetic fibers help explain the cranial autonomic symptoms [10]. The role of the hypothalamus remains unclear. Whether it triggers the attacks, has a permissive role in the self-repetition of the trigeminal autonomic reflex, or serves to terminate attacks is the subject of ongoing research [9]. It is possible that with reduced descending antinociceptive input to the brainstem, the threshold for the activation of the above-described cascade is lowered, leading to the accompanying autonomic symptoms and pain [10].

A nitric oxide–cyclic guanosine monophosphate (NO–cGMP) signaling pathway that stimulates the calcitonin gene-related peptide (CGRP) in the trigeminal ganglion has been shown to be involved in the development of PH, both in humans in vivo as well as in animal models [10,11]. Both molecules are potent vasodilators. The involvement of NO–cGMP pathways may also explain the unique effect of indomethacin. For nonsteroidal anti-inflammatory drugs (NSAIDs), cyclooxygenase 1/2 inhibition has been shown to alter the response of trigeminocervical neurons to nociceptive dural activation. In contrast to other NSAIDs, indomethacin demonstrated an additional inhibitory modulation of NO-induced vasodilation and especially of trigeminal activity [11] and the superior salivatory nucleus [12]. This knowledge also suggests the specific pathophysiology of the two indomethacin-sensitive headaches PH and hemicrania continua.

Patients with PH usually require long-term therapy [13]; only a small proportion of adult patients can discontinue indomethacin treatment without the reoccurrence of headaches [14]. Case reports about short-term treatment with indomethacin in adult and pediatric patients are rare [15,16,17]. In six of our patients (75%), the discontinuation of therapy was possible, even after 4.7 years of treatment. Nevertheless, half of these patients (*n* = 3) were able to discontinue indomethacin after less than one year. In other patients, attempts to wean them off indomethacin failed multiple times. Why indomethacin treatment could be successfully discontinued in some patients but not in others remains unclear. Some authors speculate that starting indomethacin as early in the course as possible may have affected the resolution of PH, perhaps reflecting the first course of episodic PH [14,17]. This could also explain the short course of indomethacin until successful weaning for *n* = 2 of the patients who presented. Both of these patients had symptoms for less than two months at the time of diagnosis, and the indomethacin therapy could be discontinued in both of them after less than six months.

On the other hand, there are reports of an absence of symptoms after stopping indomethacin therapy after years of treatment in adult patients with chronic PH [15]. In the other *n* = 4 patients with successful weaning from indomethacin, the time to discontinuation ranged from 0.92 to 4.7 years. In addition to a complete remission of PH under indomethacin therapy, a change in the indomethacin metabolism and an increased pain tolerance are considered contributing factors to a successful dose reduction [18].

Adverse effects appeared in *n* = 6 (75%) patients and led to the discontinuation of therapy in two patients (25%). The wide range of side effects in indomethacin therapy leads to a treatment interruption in up to 27% of adult patients [6], with gastrointestinal problems being the most common [18]. The causes of behavioral side effects such as restlessness or aggression are still subject to speculation. Possible explanations are the structural similarity of indomethacin to serotonin and a possible psychogenic effect of prostaglandin inhibition [19]. Apart from a case series in pediatric patients [2], this phenomenon has mainly been described in adult patients [19,20]. Adverse events can be partly prevented using a low maintenance dose of indomethacin. For gastric problems, proton-pump inhibitor therapy and treatment with suppositories proved helpful [2]. Due to the unique mechanism of indomethacin in the treatment of PH and the clinical observation that a long-term intake is not associated with severe side effects, the standard treatment for patients reported in this case series was the continuation of indomethacin treatment until weaning was possible without reoccurrence of PH. Changing to another medication was not considered in the absence of relevant side effects.

Alternative treatments are available for patients who cannot tolerate indomethacin due to side effects, though they are not as effective as indomethacin therapy. There are promising results for the alternate treatment with topiramate, verapamil, other NSAIDs such as celecoxib, rofecoxib, and a combination of piroxicam and β-cyclodextrin, as well as carbamazepine and amitriptyline [6,7,8]. Interventional procedures such as greater occipital nerve blocks by local injection of lidocaine and steroids [7], electrical stimulation of the optic or vagus nerve, and deep brain stimulation of the ipsilateral posterior hypothalamus remain experimental and have not been tested in children [8]. Studies on treating cluster headache patients with antibodies against the calcitonin gene-related peptide have reported cautiously positive results [8]. Nonetheless, these findings could also be new approaches for the treatment of PH. From animal models, the NO–cGMP cascade offers promising opportunities as a potential therapeutic target for PH treatment, although there is currently no specific pharmaceutical approach [11,12].

In contrast to adult patients’ management, pediatric patients with chronic PH receiving indomethacin therapy need to have their dose regularly adjusted to their body weight to maintain a sufficient treatment at the lowest maintenance dose [2]. The occasional supply shortages of indomethacin [21], the small number of suitable preparations, and the unapproved use in pediatric patients in some countries [22] also pose challenges in treating children with PH.

PH remains a rare cause of headache in children [23]. This condition highly impairs the individual’s quality of life, with a pain-related disability and frequent absences from school [2]. It took 9.5 ± 10.5 months (maximum of two years and ten months) for the patients presented here to be diagnosed and treated sufficiently, while most of them had severe headache attacks several times daily. The long period until successful therapy highlights the importance of considering PH as a differential diagnosis. Children with therapy-refractory headaches, especially those with associated neurological symptoms, should be referred to a specialized clinic.

### Limitations

Given the small and unrepresentative sample, general statements about the demographics of PH in children, statistical comparisons, or causal factors are not possible. Further studies need to determine the prevalence and incidence of PH in children and adolescents, along with factors contributing to the successful discontinuation of indomethacin.

## 5. Conclusions

To our knowledge, this is the first follow-up study to show that in pediatric patients, there is a possibility that Paroxysmal Hemicrania will cease. At this stage, we cannot provide general recommendations on the necessary treatment duration. However, knowledge about possible remission is significant for the management and counseling of patients and their parents. The discontinuation of indomethacin treatment was possible at an early stage and even after several years of therapy. Based on the available data, we suggest an early stepwise reduction to the lowest possible maintenance dose after the successful initiation of indomethacin therapy, followed by discontinuation trials every six months of treatment.

Most adverse events can be prevented using a low maintenance dose of indomethacin and proton-pump inhibitor therapy. Forthcoming research on therapeutic approaches could improve treatment options in the future.

## Figures and Tables

**Figure 1 children-08-00101-f001:**
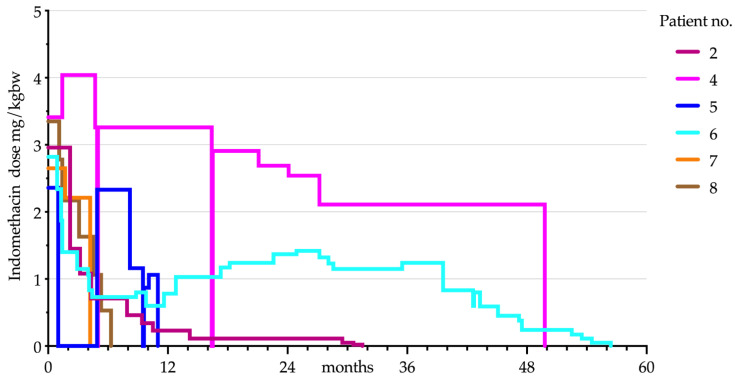
Indomethacin dose progression in relation to the respective body weight in patients who have successfully discontinued indomethacin therapy (*n* = 6).

**Figure 2 children-08-00101-f002:**
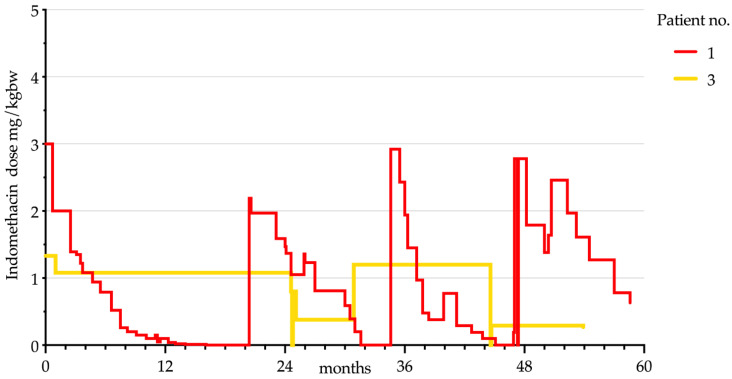
Indomethacin dose progression in relation to the respective body weight in patients still receiving indomethacin treatment (*n* = 2).

**Table 1 children-08-00101-t001:** The ICHD-3 diagnostic criteria of Paroxysmal Hemicrania.

Diagnostic Criteria
A. At least 20 attacks fulfilling criteria B–E
B. Severe unilateral orbital, supraorbital, and/or temporal pain lasting 2–30 min
C. Either or both of the following:
1. At least one of the following symptoms or signs, ipsilateral to the headache:
(a) conjunctival injection and/or lacrimation
(b) nasal congestion and/or rhinorrhea
(c) eyelid edema
(d) forehead and facial sweating
(e) miosis and/or ptosis
2. A sense of restlessness or agitation
D. Occurring with a frequency of > 5 per day ^1^
E. Prevented absolutely by therapeutic doses of indomethacin
F. Not better accounted for by another ICHD-3 diagnosis

^1^ During part, but less than half, of the active time-course of PH, attacks may be less frequent [1].

**Table 2 children-08-00101-t002:** Patient characteristics.

Item	Patient No./Gender								All Mean ± SD
	1/m	2/m	3/f	4/m	5/m	6/m	7/f	8/m	
Age at PH onset (years)	1.2	5.8	6.8	7.0	7.3	7.9	10.4	11.8	7.3 ± 3.2
Age at the time of diagnosis (years)	4.1	6.0	7.3	7.3	7.9	9.3	10.7	11..9	8.1 ± 2.5
Time to diagnosis (months)	34.5	2.5	6.3	3.3	8.0	17.7	3.7	7.7	10.5 ± 10.8
Mean pain intensity (NRS)	10	6	10	7	6	5	4	10	7.3 ± 2.5
Maximum pain intensity (NRS)	10	8	10	9	8	10	7	10	9.0 ± 1.2
Mean attack duration (min)	25	20	30	30	30	15	5	3	19.8 ± 11.1
Type of PH	EPH	CPH	CPH	CPH	EPH	CPH	CPH	EPH	CPH:EPH ratio 1.67:1
Indomethacin starting dose (mg/kgbw)	3.0	2.96	1.33	3.41	2.36	2.82	2.65	3.35	2.75 ± 0.66
Indomethacin treatment period (years)	-	2.4	-	4.2	0.9	4.6	0.4	0.5	2.2 ± 1.9

CPH: Chronic Paroxysmal Hemicrania, EPH: Episodic Paroxysmal Hemicrania, kgbw: kilogram body weight, min: minutes, NRS: numerical rating scale, 0 = no pain; 10 = worst pain (For children below the age of 11, the parents′ NRS rating has been taken into account); PH: Paroxysmal Hemicrania, SD: standard deviation. Patients no. 1 and 3 are still taking indomethacin.

**Table 3 children-08-00101-t003:** ICHD-3 diagnostic criteria for PH in the study patients.

ICHD-3 Diagnostic Criteria for PH		Patient No./Gender								All (%)
		1/m	2/m	3/f	4/m	5/m	6/m	7/f	8/m	
A	At least 20 attacks fulfilling the criteria	x	x	x	x	x	x	x	x	100
B	Severe unilateral orbital, supraorbital, and/or temporal pain lasting 2–30 min	x	x	x	x	x	x	x	x	100
C	Either or both of the following:									
1. Headache accompanied by (at least one)										
	(a) conjunctival injection and/or lacrimation				x			x		25.0
	(b) nasal congestion and/or rhinorrhea	x					x	x		37.5
	(c) eyelid edema									0
	(d) forehead and facial sweating	x	x		x	x				50.0
	(e) miosis and/or ptosis	x				x	x			37.5
2. A sense of restlessness or agitation			x	x			x	x	x	62.5
D	Occurring with a frequency of > 5 per day ^1^	x	x	x	x	x	x	x	x	100
E	Prevented absolutely by indomethacin	x	x	x	x	x	x	x	x	100
F	Not better accounted for by other diagnosis	x	x	x	x	x	x	x	x	100

^1^ During partof the active time-course of PH, attacks may be less frequent [1]. x: Fulfillment of criterion.

## Data Availability

The data presented in this study are available on request from the corresponding author. The data are not publicly available due to privacy issues.

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
