# Peer review of "Long-Term Outcome of Indomethacin Treatment in Pediatric Patients with Paroxysmal Hemicrania—A Case Series"

_children, 2021, doi:10.3390/children8020101_

Round 1
Reviewer 1 Report
Introduction
The paper presents a retrospective analysis of paroxysmal hemicrania (PH) in paediatric patients treated with indomethacin. The authors of this study report on treatment results observed in this rare disease. It is an important issue, because descriptions of PH in children and adolescents are scarce. In fact, the only case series of PH in paediatric population so far came from the same pain centre as the present study (Blankenburg et al. 2009). Moreover, there is very limited data available on long-term response to treatment.
Merits
The authors of the study provided a comprehensive presentation of a series of patients. The data provided gives satisfactory insight into the basic parameters of the studied group. Most valuably, the details regarding indomethacin dosing over time have been presented.
Critique
MAJOR
None
MINOR
- L17 “Indomethacin represents both a diagnostic criterion and the treatment of choice […]” Not indomethacin per se, but its effectiveness is a diagnostic criterion.
- L33 Trigeminal autonomic cephalalgias (TACs) are not a ‘condition’, but a group of disorders.
- L71-75 It is unclear whether patients with probable PH according to International Classification of Headache Disorders were excluded from this case series.
- Table 2. It is unclear why mean time to diagnosis was over 10 months, when the difference between age of PH onset (7.5) and age of diagnosis (7.9) was only 0.4 years.
- Table 2. Are the authors convinced that all subjects had their pain assessed with numerical rating scale (NRS)? How was it performed in a 3-year-old subject? Perhaps some behavioural pain scales were employed?
- The present study is probably based on the same cohort as a previous study from this group (Blankenburg et al. 2009). The numbers are similar (n = 8). Are the characteristics given in the 2009 study valid also for this group, or have some new patients been introduced? If new patients are described, than their data could be included e.g. in supplementary materials. This is an important issue since less than 30 cases of PH in paediatric population have been described to this day. Moreover, the data provided should involve some comment on the classification of PH in this case series. How many patients were diagnosed with chronic or episodic PH? If such a distinction was impossible to perform (e.g. due to constant administration of indomethacin), it should be also specified.
- L 88 abbreviation kgbw should be explained (e.g. ‘kg body weight’).
- L 104 Since it is a study concentrating on indomethacin, serious adverse effects (SAE) must be comprehensively described, especially if they led to treatment discontinuation. The 2009 study does not provide data on any severe complications according to current definitions of SAE.
- L 106 Some comment regarding topiramate effectiveness should be given: what were the effective doses? What were the effects of treatment except for headache remission observed for 2 years? Perhaps this data was provided in the 2009 paper?
- L136 ‘salivatory’ (spelling mistake)
- L230 Authors should reconsider whether indomethacin weaning should not be encouraged ‘every’ six months, instead of just once ‘after’ six months.
Author Response
Reviewer #1
Introduction
The paper presents a retrospective analysis of paroxysmal hemicrania (PH) in paediatric patients treated with indomethacin. The authors of this study report on treatment results observed in this rare disease. It is an important issue, because descriptions of PH in children and adolescents are scarce. In fact, the only case series of PH in paediatric population so far came from the same pain centre as the present study (Blankenburg et al. 2009). Moreover, there is very limited data available on long-term response to treatment.
Thank you for your review. We are pleased that you enjoyed the article on this rare but severe disease. Your comments allow us to make further improvements to the text.
Merits
The authors of the study provided a comprehensive presentation of a series of patients. The data provided gives satisfactory insight into the basic parameters of the studied group. Most valuably, the details regarding indomethacin dosing over time have been presented.
Critique
- L17 “Indomethacin represents both a diagnostic criterion and the treatment of choice […]” Not indomethacin per se, but its effectiveness is a diagnostic criterion.
Thank you for the comment. We have now been more specific in the abstract.
- L33 Trigeminal autonomic cephalalgias (TACs) are not a ‘condition’, but a group of disorders.
Thank you for the advice. We have now been more precise in the introduction.
- L71-75 It is unclear whether patients with probable PH according to International Classification of Headache Disorders were excluded from this case series.
Thank you for this helpful advice. We included 2 tables with the patient characteristics (table 2 and 3). All patients fulfilled the ICHD-3 criteria for either EPH or CPH. No patient in our case series had a probable PH. Two patients met ICHD-3 criteria by the new diagnostic criterion of "A sense of restlessness or agitation."
- Table 2. It is unclear why mean time to diagnosis was over 10 months, when the difference between age of PH onset (7.5) and age of diagnosis (7.9) was only 0.4 years.
Thank you for the important comment. We made a transfer error here when evaluating the age at onset of PH. We have corrected this in table 2 and the Results section.
- Table 2. Are the authors convinced that all subjects had their pain assessed with numerical rating scale (NRS)? How was it performed in a 3-year-old subject? Perhaps some behavioural pain scales were employed?
Thanks for this advice. For children below the age of 11, the parents' NRS rating has been taken into account. We have added this information in the corresponding passage of the results.
- The present study is probably based on the same cohort as a previous study from this group (Blankenburg et al. 2009). The numbers are similar (n = 8). Are the characteristics given in the 2009 study valid also for this group, or have some new patients been introduced? If new patients are described, than their data could be included e.g. in supplementary materials. This is an important issue since less than 30 cases of PH in paediatric population have been described to this day. Moreover, the data provided should involve some comment on the classification of PH in this case series. How many patients were diagnosed with chronic or episodic PH? If such a distinction was impossible to perform (e.g. due to constant administration of indomethacin), it should be also specified.
Thank you for the remark. We have summarized the information on the individual patients in the revised Table 2. The applicability of the ICHD-3 criteria to the study patients can be found in the new Table 3.
- L 88 abbreviation kgbw should be explained (e.g. ‘kg body weight’).
Thank you for the feedback. We have now added an explanation for the abbreviation at the first occurrence in the text.
- L 104 Since it is a study concentrating on indomethacin, serious adverse effects (SAE) must be comprehensively described, especially if they led to treatment discontinuation. The 2009 study does not provide data on any severe complications according to current definitions of SAE.
Thank you for pointing this out. Fortunately, none of the patients suffered from a serious adverse effect as defined by e.g. clinicaltrials.gov „…death, a life-threatening adverse event, inpatient hospitalization or prolongation of existing hospitalization, a persistent or significant incapacity or substantial disruption of the ability to conduct normal function…” (https://prsinfo.clinicaltrials.gov/results_definitions.html#AdverseEventsDefinition). We included this information in the Results section.
- L 106 Some comment regarding topiramate effectiveness should be given: what were the effective doses? What were the effects of treatment except for headache remission observed for 2 years? Perhaps this data was provided in the 2009 paper?
Thank you for the note. We have added the information on the effective dose of topiramate in the corresponding line. Since both patients did not return to the outpatient pain clinic after conversion to topiramate, we are unable to comment on other effects. These were also not questioned in the interview.
- L136 ‘salivatory’ (spelling mistake)
Many thanks for the comment! We have corrected the corresponding passage.
- L230 Authors should reconsider whether indomethacin weaning should not be encouraged ‘every’ six months, instead of just once ‘after’ six months.
Thank you for the good suggestion. We have revised the passage in the Conclusion section. This will help us to make the recommendation for our treatment approach clearer.
Reviewer 2 Report
Overall, this is a well-written manuscript on an uncommon disorder. Please see the recommendations below:
-In the introduction, I recommend emphasizing that a distinguishing feature of paroxysmal hemicrania from other TACs is the length of attacks.
-For an 8 patient case series, a table with each patients characteristics would be very helpful in summarizing the case data (ex. age, sex, duration of symptoms/disease, indomethacin dosing quantity and duration, etc).
-While this was a small sample due to the rarity of the disorder in the pediatric population, do you feel the demographics reflect those in the literature for this disorder? PH tends to have a higher ratio of female patients than the other TACs, yet your sample had more males.
-Did patient's try any alternative agents to help them taper off the indomethacin? We commonly try to transition patients to another agent more appropriate for long-term use to avoid complications from indomethacin therapy. If this was not attempted, please comment on why not.
-Did patient demographics (age, sex) seem to have any impact on disease course and/or ability to wean indomethacin?
Author Response
Reviewer # 2
Overall, this is a well-written manuscript on an uncommon disorder. Please see the recommendations below:
Thank you for your review. We are pleased that you enjoyed the article. Your review allows us to make further improvements to the text.
-In the introduction, I recommend emphasizing that a distinguishing feature of paroxysmal hemicrania from other TACs is the length of attacks.
Thank you for the comment. We have included this unique feature in the abstract and in the introduction.
-For an 8 patient case series, a table with each patients characteristics would be very helpful in summarizing the case data (ex. age, sex, duration of symptoms/disease, indomethacin dosing quantity and duration, etc).
Thank you for the remark. We have summarized the information on the individual patients in the revised Table 2. The applicability of the ICHD-3 criteria to the study patients can be found in the new Table 3.
-While this was a small sample due to the rarity of the disorder in the pediatric population, do you feel the demographics reflect those in the literature for this disorder? PH tends to have a higher ratio of female patients than the other TACs, yet your sample had more males.
Given the small number of patients and the nonrepresentative sample of the Paediatric Pain Centre, we unfortunately cannot make any deductions about the demographics of PH in children. We’ve emphasized this in the Limitations section.
-Did patient's try any alternative agents to help them taper off the indomethacin? We commonly try to transition patients to another agent more appropriate for long-term use to avoid complications from indomethacin therapy. If this was not attempted, please comment on why not.
Thank you for bringing up this interesting question. For patients with a suspected diagnosis of PH, we only use indomethacin. Because of its unique mechanism of action, we then also continue the indomethacin therapy in this way after a successful therapy trial. We have added this information in the Discussion section.
-Did patient demographics (age, sex) seem to have any impact on disease course and/or ability to wean indomethacin?
Unfortunately, apart from the vague suggestion that indomethacin may be discontinued earlier if the duration of illness is short, we cannot make any further deductions because of the small sample.